# Aero-Engine Blade Cryogenic Cooling Milling Deformation Simulation and Process Parameter Optimization

**DOI:** 10.3390/ma16114072

**Published:** 2023-05-30

**Authors:** Ting Chen, Yun Xu, Bo Huang, Yan Shi, Jiahu Zhang, Lei Li, Yaozhi Meng, Xuqing Li

**Affiliations:** School of Mechanical Engineering, Sichuan University of Science and Engineering, Zigong 643000, China; 321085503303@stu.suse.edu.cn (T.C.); xuyunzigong@163.com (Y.X.);

**Keywords:** aero-engine blade, titanium, cryogenic machining, simulations, particle swarm algorithm

## Abstract

For the machining of aero-engine blades, factors such as machining residual stress, milling force, and heat deformation can result in poor blade profile accuracy. To address this issue, simulations of blade milling were completed using DEFORM11.0 and ABAQUS2020 software to analyze blade deformation under heat-force fields. Process parameters such as spindle speed, feed per tooth, depth of cut, and jet temperature are used to design both a single-factor control and BBD test scheme to study the influence of jet temperature and multiple changes in process parameters on blade deformation. The multiple quadratic regression method was applied to establish a mathematical model correlating blade deformation with process parameters, and a preferred set of process parameters was obtained through the particle swarm algorithm. Results from the single-factor test indicated that blade deformation rates were reduced by more than 31.36% in low-temperature milling (−190 °C to −10 °C) compared with dry milling (10 °C to 20 °C). However, the margin of the blade profile exceeded the permissible range (±50 µm); therefore, the particle swarm optimization algorithm was used to optimize machining process parameters, resulting in a maximum deformation of 0.0396 mm when the blade temperature was −160 °C~−180 °C, meeting the allowable blade profile deformation error.

## 1. Introduction

The aero-engine blade is an important part of the aero-engine. Manufacturing materials generally include high-strength stainless steel, titanium alloy, and high-temperature alloys which possess high strength, good heat resistance, and corrosion resistance. However, these materials have a large elastic modulus and poor thermal conductivity. During the milling process, heat generated cannot be conducted out in time, causing accelerated tool wear. Increased milling force and heat generation can also impact the blade processing accuracy and reduce surface quality, resulting in inconsistency between actual processing parameters and design values.

Cryogenic machining has been studied for its effectiveness to enhance surface integrity, reducing tool wear and improving accuracy and productivity. It is considered a green manufacturing technology to replace traditional cutting fluids while also aligning with the latest trends in global sustainable manufacturing. At present, low-temperature machining studies focus on the cutting mechanism, temperature, force, tool wear, and surface integrity. For example, Bejjani R et al. [1] investigated the effect of changing the nozzle position on the deep cooling process using the finite element method, CFD method, and experimental methods. Shokrani A et al. [2] investigated the most suitable tool geometry for low-temperature machining, which significantly affected the finishing performance of Ti-6Al-4V, increasing the material removal rate; Hong S Y et al. [3] directed LN_2_ through microjets at the flanks, front edge, or sides near the cutting edge to study its effects. The cutting temperature was theoretically estimated using the finite element method to compare low-temperature machining with conventional dry cutting and emulsion cooling. Yin X et al. [4] investigated low-temperature large-strain extrusion machining (CT-LSEM) as a novel severe plastic deformation (SPD) method for producing ultrafine-grain (UFG) microstructures. Shokrani A et al. [5] investigated the effect of liquid nitrogen cryogenic cooling on the surface integrity of end milling of Ti-6Al-4V titanium alloy. Thamizhmanii S et al. [6] used machining materials with superhard tools such as cubic boron nitride (CBN) and poly cubic boron nitride (PCBN) to reduce tool wear while achieving dimensional accuracy, smooth surfaces, and more parts per cutting edge. Kaynak Y et al. [7] investigated the machining performance of a new generation of near-beta alloy Ti-5553 for high-speed cutting and dry machining under low-temperature cooling conditions, which resulted in shorter chips compared to dry machining. Dai X et al. [8] investigated the effect of dry and low-temperature machining conditions on the machining hardened layer at different cutting speeds based on the machining of CrNiFe alloy. Iqbal A et al. [9] quantified the sustainability of milling a commonly used titanium alloy (Ti-6Al-4V) by varying the mass flow rate of two cryogenic coolants at different cutting speeds. Three cooling methods were tested: dry (no coolant), evaporative cryo coolant (liquid nitrogen), and throttled cryo coolant (compressed carbon dioxide gas). M. Ukamanal et al. [10] studied the machining performance of AISI 316 stainless steel in dry and spray impingement cooling environments, compared to dry processing; machining in a spray impingement cooling environment is most effective for turning AISI 316 stainless steel, with over a 300% reduction in chip temperature. Danish M et al. [11] studied machining under dry and low-temperature cooling conditions by varying the cutting speed, feed rate, and depth of cut, and developed a prediction model for surface roughness under different machining environments. Liu J Y et al. [12] investigated the cutting force, cutting temperature, and tool life of high-speed steels in drilling nickel-based superalloy Inconel 718 through orthogonal cutting experiments. Then the empirical method of multi-factor line regression was used to establish the empirical models of drilling torque, thrust force, cutting temperature, and tool life of the drills with emulsion as the cutting fluid, and to verify the validity of the established model, and the significance of the regression method is tested to verify the effectiveness of the model. Caudill J et al. [13] developed a three-dimensional finite element model for milling Ti-6Al-4V under low temperature and a cutting fluid environment was developed based on Deform, which predicted cutting forces and temperature fields during machining, with results showing that milling at low temperatures can extend tool life and improve machining efficiency. Boulila A et al. [14], according to AISI 304 stainless steel blade milling process, studied the effects of the cutting parameters on tool deflection, tracking errors, and surface roughness; the results have shown how much the tool deflection and the tracking errors were mainly influenced by the axial depth and the cutting speed.

Jamil M et al. [15] compared low-temperature CO_2_ and hybrid nanofluid-based MQL technologies to study the effects of both cooling and lubrication techniques on turning Ti-6Al-4V. The results showed that the hybrid nano-additive resulted in an 8.72% reduction in average surface roughness, an 11.8% reduction in cutting forces, and a 23% increase in tool life compared to low-temperature cooling. However, the cryogenic technique showed an 11.2% reduction in cutting temperature compared to MQL-hybrid nanofluid at low and high cutting speed and feed rate levels. Mkg A et al. [16] used AdvantEdge7.3 software to set up predictive models to evaluate the cutting forces and cutting temperatures when machining AA2024-T351 alloy under dry, liquid nitrogen (LN_2_), and carbon dioxide (CO_2_) conditions. The results showed that the results obtained from the simulation model were very close to the experimental results, which verified the validity of the finite element simulation. Gupta M et al. [17] investigated that under sustainable cooling conditions, compared to Ranque-Hilsch Vortex tube plus MQL (RHVT + MQL), liquid nitrogen plus minimum quantity lubrication (N_2_ + MQL) and liquid nitrogen (N_2_), N_2_ + MQL achieve the minimum specific cutting energy and surface hardness. Danish M et al. [18] demonstrated that by turning Inconel 718, LN_2_-aided machining significantly lowers total machining costs and energy consumption as compared to dry, MQL, cryogenic-CO_2_ conditions. Furthermore, when compared to dry processing circumstances, the LN_2_ cooling environment can significantly reduce processing output, assisting in the sustainability of the aerospace sector. Korkmaz M E et al. [19] performed the turning experiments under dry, minimum quantity lubrication (MQL different positions), and nano-MQL (different positions) conditions Nimonic 80A, and the influence of nozzle position during MQL and nano-MQL was investigated. The results show that the total tool wear is approximately 60% better for nano-MQL (mixed direction) than in dry conditions. Airao J et al. [20] investigated the machinability of Ti-6Al-4V in conventional and ultrasonic assisted turning (UAT) under dry, wet, MQL, and LCO_2_ conditions. The findings of ultrasonic-assisted turning reveal that LCO_2_ and ultrasonic vibration greatly reduce specific cutting energy while maintaining surface roughness and tool life. Gupta M K et al. [21] conducted an experimental study to put forward a comparison of tool wear mechanism under dry, cryogenic-LN_2_, MQL, and hybrid LN_2_-MQL cooling conditions in the turning of titanium alloy. The SEM and EDS analysis is carried out to analyze the tool wear under different cooling/lubrication environments. Pereira O et al. [22] presented cryogenic cooling with external MQL lubrication (CryoMQL) working along with CO_2_ as an internal coolant for milling Inconel 718. The results demonstrate that internal CryoMQL enhances tool life by 57% when compared to emulsion coolant, and 120% when compared to MQL in stand-alone mode. Under dry and MQL circumstances, the Xu J et al. [23] conducted experimental research on the drilling behavior and hole quality of aerospace-grade CFRP/Ti-6Al-4V sandwiches using TiAlN-coated and diamond-coated drills. Leksycki K et al. [24] studied the influence of cutting parameters on the surface texture of Ti-6Al-4V titanium alloy when finishing turning under dry and wet cooling conditions. Khanna N et al. [25] established eco-friendly cutting fluid techniques for turning AZ91/5SiC, unique setups of minimal quantity lubrication (MQL), cryogenic, and CryoMQL machining with LN_2_. CryoMQL technology has 25.59% and 18.35% lower Ra values when compared to MQL and cryogenic machining, respectively. Khanna N et al. [26] carried out a study to analyze the dry, flood, MQL (minimum quantity lubrication), and cryogenic machining with LCO_2_ in terms of tool wear for the turning tests of 15-5 PH SS. Chetan et al. [27] compared techniques such as cryogenic cooling and cryogenic treatment with the Al_2_O_3_ nanoparticles based MQL cooling method (nMQL) in turning off the nickel-based Nimonic 90 alloy. The cryogenic cooling environment has been found to be the best mode for the machining of the Nimonic 90 alloy. Jebaraj M et al. [28] examined the performance and the influence of carbon dioxide (CO_2_) and liquid nitrogen (LN_2_) on cutting zone temperature (T_c_), feed (F_x_), normal (F_y_), and thrust (F_z_) forces, tool wear, chip morphology, surface morphology, and roughness (R_a_) in the machining of die steel. Varghese V et al. [29] carried out a study where the tungsten carbide cobalt inserts are cryogenically treated (CT) for 24 h at a soaking temperature of 195.8 °C and utilized to test milling performance in dry, wet, and cryogenic cutting regimes. Wika P et al. [30] investigated the effect of supercritical carbon dioxide cooling with minimum quantity lubrication (scCO_2_ + MQL) on tool wear and surface integrity of AISI 304 L austenitic stainless steel in milling.

The above literature studies the properties of low-temperature processing and describes the effects of various cooling methods on parameters such as surface roughness, temperature, tool wear, tool life, and productivity. However, the theory of designing low-temperature milling parameters for different materials or parts is not yet mature; in particular, low-temperature milling of thin-walled parts with complex surfaces is studied.

We aimed at the design of low-temperature milling parameters of aviation blades, based on the simulation test method. A single-factor test was used to determine the relationship between milling force, milling temperature, and blade deformation. Using the BBD pilot program and Design-Expert12.0 data analysis software, the relationship between process parameters and blade deformation was obtained using the best binary polynomial regression model. By considering blade deformation (less than ±50 µm) as the objective function and process parameters (i.e., spindle speed, feed per tooth, depth of cut, and jet temperature) as constraints, the preferred set of process parameters was determined using the particle swarm optimization algorithm.

## 2. Material and Methods

### 2.1. Low-Temperature Jet Principle and Device

The structure of an aero-engine blade is complex, and the non-linear and time-varying cutting thickness during high-speed milling results in blade elastic machining deformation. Dry milling accelerates tool wear; for the thin-walled blade, by cutting, the heat increases, which accelerates the blade elastoplastic bending torsional deformation; a low temperature is achieved with jet pressure into the tool cutting area as a cooling medium; gasification occurs for titanium alloy and surrounding gas through convective heat exchange so that the cutting heat with high-speed milling is linearly reduced. This paper adopts external pipeline low-temperature jet cooling and sets up a pipeline outside the processing equipment to transport and round the jet low-temperature coolant, as shown in Figure 1. To ensure adequate cooling of the tool, turbulent jets are generally used, and axisymmetric free jets. *a* is the expansion angle, the jet diffusion angle α is seen in Equation (1). A low-temperature coolant with a pressure of 0.08 MPa and a flow rate of 4 L/min cools the tool intending to reduce the temperature of the cutting area.
(1)tgα=3.4a

### 2.2. Tool-Workpiece Finite Element Model Building

In this paper, the tool is treated as a rigid body and the workpiece as a plastic body. High temperature, large deformation, and large strain rate in the actual machining can cause mesh distortion and non-convergence of the titanium alloy material during the simulation process. To address this issue, an automatic re-division and mesh window density function is used to encrypt the mesh locally for the workpiece, thus improving the simulation accuracy. After meshing, the number of tool meshes is 45,868 with a window density ratio of 0.01 and the number of workpiece meshes is 111,009 with a window density ratio of 0.001. The resulting tool-workpiece model after meshing is shown in Figure 2.

By utilizing the UG/CAM module, the blade machining path was prepared using a helical milling strategy, as depicted in Figure 3. The process knife axis remained perpendicular to the blade surface with a lateral tilt angle of 90° and a forward tilt angle of 0°. From the literature [31], it is known that the maximum deformation occurs at the middle part of the leaf body between 40–50 mm, and the bending deformation tends to decrease from the middle part of the leaf body to the root and the tip ends of the leaf. Therefore, this paper selects a tool path at the middle section of the leaf body with 46 tool contacts. Milling force was applied along the normal direction to the nodes of the blade surface grid nearest to the tool contacts. The node coordinate system was adjusted to be consistent with the outer normal of the surface, followed by the application of a normal force at the node to simulate the blade milling process, as shown in Figure 4.

### 2.3. Settings of Physical Parameters

A numerical model of local dynamic milling of the blade is established. Currently, the effect of the clamping position of the fixture is not considered, and the milling width is constant at 2 mm. The tool material used is a WC-based carbide ball end mill (YG8), with its geometric parameters shown in Table 1, and physical parameters shown in Table 2. The blade material used is TC4, with its physical parameters [32] shown in Table 3.

### 2.4. Material Ontogeny Equations

During the actual milling process of TC4 alloy material, the milling force causes high temperature, strain, and a large strain rate to produce elastic-plastic strain in the workpiece material. In this paper, we combine the effects of the elastic-plastic flow of the workpiece material during milling and the combined factors such as strain, strain rate, and milling heat that contribute to material hardening. A Johnson–Cook principal structure model [33] function was developed as shown in Equation (2).
(2)$$\sigma =\left[A+B_{\varepsilon }^{n}\right]\left[1+c\ln\left(\frac{\dot{\varepsilon }}{{\dot{\varepsilon }}_{0}}\right)\right]\left[1-{\left(\frac{T-T_{r}}{T_{m}-T_{r}}\right)}^{m}\right]$$
where *A*, *B*, *c*, *n*, *m* are material constants, $T_{m}$ is the material melting point, $T_{r}$ is the room temperature, and ε is the strain.

The Johnson–Cook model has been applied numerous times to titanium alloy materials at high temperatures, but it has not been tested under low-temperature conditions. However, the heat field inside the workpiece results in the deformation of the blade generated by the heat transfer phenomenon, which is a relatively slow process and does not fall below zero. The temperature in the deformation region is not lower than 300 °C. The above J–C model in literature [34,35,36,37,38,39,40] has been extensively verified. In addition, the temperature in the deformation region workpiece is not lower than the initial temperature. Therefore, the J–C model does not need to be calibrated for low-temperature regions. The parameters of the reference strain Johnson–Cook model are shown in Table 4.

During the milling process, pressure and friction between the tool and the workpiece (chip) result in the non-uniform distribution of temperature and pressure and different contact and friction characteristics. Therefore, a reasonable relationship between the contact-friction-thermal characteristics is established. The study focuses on the contact relationship between the tool-workpiece and workpiece-chip, with the tool being the primary object and the workpiece as the secondary object. The Zorev friction model is used for the friction between the tool and chip, and its contact area is divided into bonding and sliding zones. In the bonded region, shear stress equals the shear yield strength of the material. While in the sliding region, the relationship between the tool and the chip satisfies Coulomb’s law of friction, as shown in Equation (3).
(3)$$\tau_{f}=\left\{\begin{array}{l} \mu \sigma_{n},\sigma_{n}<\tau_{s}\;\;\;\;(Zone\;of\;sliding\;friction)\\ \tau_{s},\mu \sigma_{n}\geq \tau_{s}\;\;\;\;\;(Bonded\;friction\;zone) \end{array}\right.$$
where, $\tau_{f}$ is the tool-chip friction stress; μ is the tool-chip friction coefficient; $\sigma_{n}$ is the tool-chip contact stress; $\tau_{s}$ is the workpiece material shear flow stress. In this paper, the friction coefficient μ is set to a fixed value of 0.35.

We set up the adiabatic exchange between the blade and the fixture restraint, as well as heat exchange with the environment modeled by heat transfer coefficient (HTC), which is set to HTC = 500 dry milling (as shown in Equation (4)). $q_{c}$ is the heat flow $T_{surf}$ is the blade surface temperature, $T_{coolant}$ is the contact zone temperature. This test uses low temperatures to cool the tool surface, with both ambient and blade surface temperatures being set to 20 °C. As the chip contact surface is cooled by the nozzle, the tool surface temperature is considered to be −196 °C. Surface temperature-dependent heat transfer coefficients HTC are shown in Table 5.
(4)$$q_{c}=HTC(T_{surf}-T_{coolant})$$

When contact is in the forced sticking region, where the tool is rigid, heat exchange occurs between the blade and tool, as shown in Equation (5). According to the heat distribution coefficient (HPC), β, $q_{t}$ heat flow sharing for tools, $q_{wp}$ and heat flow shared for the workpiece (see Equation (6)), the heat flow $q_{f}$ due to friction can be calculated from the sliding speed $v_{sl}$ (see Equation (7)). When calculating HPC, the accumulation of fluid can be based on every single object $e=\sqrt{\rho kc_{p}}$, where ρ is the density, k is the thermal conductivity, and $c_{p}$ is the specific heat capacity. In this paper [42], $e_{t}=\sqrt{\rho_{t}k_{t}c_{pt}}=1111\frac{w}{k}\sqrt{\frac{s}{m}}$.
(5)$$\begin{array}{l} q_{t}=\beta q_{f}\\ \;\;=\frac{e_{t}}{e_{t}+\sqrt{\rho_{w}k_{w}c_{pw}}}q_{f} \end{array}$$
(6)$$q_{wp}=\left(1-\beta \right)q_{f}$$
(7)$$q_{f}=\tau v_{sl}$$

## 3. Blade Milling Deformation Simulation Test and Data Analysis

### 3.1. Milling Force and Milling Temperature Data Processing

To obtain load data on the blade subjected to milling forces and thermal deformation, a milling simulation of the blade localization is required. Based on the established 3D milling numerical model, the initial machining process parameters are selected. After assembling the completed model in UG using the process parameters, the geometric model is exported in the corresponding format of deform and checked for accuracy. Since accurate data output is required, incremental steps should be taken to meet the values for which valid data can be output. In addition, the tool’s initial time limit for machining the workpiece is limited to seconds to observe the pattern. The three-dimensional milling key performance data are formulated for numerical simulation to calculate the three-way milling force and milling temperature during blade milling, and the preliminary process parameters and numerical simulation key performance parameters are shown in Table 6 and Table 7.

A graph of three-way milling force and milling temperature versus machining time was generated using the DEFORM post-processing module. However, the exported data exhibited high frequency and noise, leading to inaccuracies in the fetch points. Therefore, a polynomial fitting function in the origin software was used to perform multi-order fitting of the three-way milling force and milling temperature data. The fitted curve plots are shown in Figure 5 and Figure 6.

At the beginning of tool insertion into the workpiece, the material is in the high strain, high strain rate stage before the thermal softening stage occurs. As a result, the temperature of the workpiece and the milling force fluctuate widely. Thermal softening of the material occurs after 0.03 s, the milling force and milling temperature fluctuate up and down around a fixed value, and the cutting force distribution is consistent with that of Albertelli P [43]. To obtain a relatively more accurate three-way milling force and milling temperature, the maximum value is taken in the period of 0.05 s~0.1 s (as in Figure 7). The resulting load data is then used to calculate blade deformation, as outlined in Table 8.

As can be seen from the data in Table 8, the F_y_ values were greater than F_x_ and F_z_ during the milling process, in agreement with the experimental results obtained by Shi Y [44]. Therefore, the model for blade milling thermal-force data acquisition is feasible. However, the study of blade deformation based on milling forces and heat alone is insufficient, and further investigation into blade deformation simulation is needed.

### 3.2. Blade Deformation Data Processing

To further investigate the deformation of each tool contact during blade milling, the acquired three-way milling force and milling temperature data were applied to the key tool contacts of the blade local coordinate system using load based on ABAQUS 2020 simulation software. After the simulation was completed, the software post-processing interface was entered, and cell node data was extracted using the field output command. The maximum blade deformation output for each tool contact during a tool walk was then obtained, as shown in Figure 8 and Figure 9.

According to Figure 8, the displacement of blade node coalescence is greatest in the middle of the leaf shape, gradually decreasing distortion from the center to both ends. This is due to the blade ends being fully constrained, the displacement being zero, the intake and exhaust sides being in the overhang position, and the thickness of the curved edge of the leaf edge being small in comparison to the thickness of the middle position, making it susceptible to deformation due to a three-way milling force and milling heat. Figure 9 illustrates that during machining with the tool feed (load moving action), the amount of blade cross-section local tool contact deformation displays a “W” shape law. In the test, the milling path of 190.4 mm equates to a maximum blade deformation of 0.6145 mm. According to the research, the inlet and exhaust sides of the blades are more prone to deformation than the middle of the back and inner airway arcs. Therefore, this paper selects the milling path of 190.4 mm near the middle maximum for studying blade machining deformation control and process parameters optimization.

### 3.3. Experimental Program

In this test, the single-factor test and Box–Benhnken test protocols were used, respectively. The single-factor test can reflect in detail and intuitively the trend and change pattern of the influence of the parameter on all the investigated targets. However, the test volume of this method is large, and the combined effect of each factor on the test index cannot be fully summarized. On the other hand, the Box–Benhnken test protocol reduces the number of trials, reduces test cycle time, and quickly finds the optimal combination of process parameters and the best process solution. This makes up for the shortcomings caused by single-factor tests.

### 3.4. Single-Factor Test Results and Analysis

There is multi-physical field action during blade milling. Among them, the milling force is the main factor affecting the profile accuracy. To study the change law of the tool-workpiece milling force at different temperature milling, we had to maintain the process parameters of spindle speed n, feed per tooth $f_{z}$, and depth of cut $a_{p}$. The levels of the selected jet temperature are −190 °C, −130 °C, −70 °C, −10 °C, and 50 °C, respectively. Details are shown in Table 9.

Numerical simulation by substituting the parameter set of the jet temperature single-factor test scheme into the finite element simulation model. The effect law of different horizontal jet temperatures on the three-way milling force, milling temperature, and blade deformation was obtained, as shown in Figure 10, Figure 11 and Figure 12.

From the above trends in Figure 10, Figure 11 and Figure 12, the trend of increasing milling force and milling temperature in three directions at 50 °C is obvious in the single-factor test (F_x_ = 76 N, F_y_ = 130 N, F_z_ = 25 N, milling temperature 300 °C). The slope of F_y_ with respect to F_x_ and F_z_ is maximum at this temperature. This can be attributed to heat transfer from the tool resulting in lower temperatures during the milling process and an increase in titanium alloy flow stress, which strengthens the material and reduces damage. Within the range of −190 °C~−10 °C, F_x_, F_y_, F_z_, and milling temperature have a small range. However, there is a continuous trend of increasing with the jet temperature. Moreover, the blade deformation trend is always consistent with the trend of milling force and milling temperature in three directions. Blade deformation reaches 0.8134 mm at 50 °C, while the minimum blade deformation is 0.6121 mm, and the jet temperature is −10 °C. Due to the small compressive strength at −70 °C, for the high frequency of fluctuation of milling force and milling temperature in the machining area, the maximum values are F_x_ = 67 N, F_y_ = 121 N, F_z_ = 10 N, and milling temperature 242 °C, respectively, resulting in an abnormal increase in blade deformation. The test proves that −190 °C is not the best temperature for controlling the deformation of the blade. Although the range of blade deformation variation is reduced relative to dry milling, the blade profile accuracy exceeds 50 µm. Therefore, using low-temperature processing, the scrap rate is still high. In this paper, the particle swarm algorithm is used to optimize for this problem and improve blade machining accuracy.

### 3.5. Box–Benhnken Test Analysis and Results

There are many process parameters that affect blade machining deformation, including spindle speed, milling speed, milling depth, milling width, feed per tooth, tool rotation angle, and more. Each parameter has a wide range of values, and the full-scale test method requires a large workload. The Box–Benhnken test scheme can achieve the same results as the full-scale test method with fewer tests, which improves testing efficiency.

This study took four elements into account based on actual operating conditions: spindle speed n, feed per tooth $f_{z}$, depth of cut $a_{p}$, and jet temperature t. Each factor is assigned three levels. The influence law of process parameters on blade deformation was explored using the Box–Benhnken test, and the ideal combination of factor values was determined using ANOVA. As a result, the workpiece has outstanding surface quality, high machining accuracy, and high machining efficiency. Table 10 shows the level values for each element depending on processing experience and experimental processing settings.

According to the actual working conditions, four factors are the subject of this paper. Therefore, it is designed for quaternion quadratic regression combination. Response surface center composite design selection using Design-Expert data analysis software.

According to the designed experimental protocol, each group has a different combination of milling parameters, uses UG10.0 software to build a 3D milling model according to each parameter combination and complete the milling simulation of this model in Deform, and outputs three-way milling force and milling temperature data. ABAQUS-based deformation simulation with milling force and milling temperature data is added as loads to the blade surface, observing blade deformation in the post-processing interface after the calculation is completed, and importing the obtained deformation data into Origin for data processing to obtain usable data and extract blade deformation data. The test result data are shown in Table 11.

From the table, the deformation value of the blade values ranges from 31.80 µm to 1005.89 µm, with 70% of the test program group exceeding the blade profile tolerance ±50 µm. This is extremely detrimental to the manufacturing quality of the blades. To investigate the potential pattern between the co-variation of multi-factor process parameters and the amount of blade deformation, further analysis of the data in the pilot program group is needed.

## 4. Optimize Process Parameters to Control Blade Deformation

### 4.1. Establish a Multivariate Nonlinear Regression Model between Process Parameters—Blade Deformation

The response surface method can reflect the relationship between response value and response factor, overcoming the defect that the orthogonal test cannot give a visual graph. The hypothetical influence factor significance level is *p* < 0.05. Analysis of the relationship between milling force components and milling parameters was carried out using the response surface center composite method. A second-order regression equation model was fitted using the least squares method based on the experimental data. The analysis of variance (ANOVA) was performed to determine the goodness of fit of the model. The second-order mathematical model is shown in Equation (8).
(8)$$y=\beta_{0}+\sum_{i=1}^{k}\beta_{i}x_{i}+\sum_{i=1}^{k}\beta_{ii}x_{i}^{2}+\sum_{i<j}^{k}\beta_{ij}x_{i}x_{j}$$
where *y* is the response value and *x* is the milling factor variable, $\beta_{0}$, $\beta_{i}$, $\beta_{ii}$, $\beta_{ij}$ are the respective coefficients.

Design-Expert data analysis software was used to perform multiple regression fitting of experimental data, representing the linear and non-linear effects of single factors and the interaction effects between multiple factors. A multivariate quadratic polynomial was used in this study, and the regression model coefficients and significance test results are presented in Table 12. The quadratic multinomial regression model is obtained in Equation (9).
(9)$$\begin{array}{l} T_{max}=0.2632+0.1462\times n+0.2765\times f_{z}-0.0655\times a_{p}+0.2819\times t+0.0682\times n\times f_{z}\\ \;\;\;\;\;\;\;\;-0.1358\times n\times a_{p}-0.0443\times n\times t-0.0528\times f_{z}\times a_{p}-0.0538\times f_{z}\times t-0.2185\times a_{p}\times t\\ \;\;\;\;\;\;\;\;+0.01098\times n^{2}+0.1228\times f_{z}^{2}+0.1452\times a_{p}^{2}-0.8724\times t^{2} \end{array}$$

From the table, the model F-value is 6.54 and the *p*-value is less than 0.001, showing that the model is highly significant. Therefore, B and D in this model have a greater degree of influence on the model. According to the F value, it is clear that the jet temperature has the greatest effect on the amount of blade deformation (F value of 29.63), followed by feed per tooth (F value of 13.23). The minimal effect of spindle speed on deformation, missing items, is less than 0.0001.

### 4.2. Particle Swarm Algorithm Modeling

The particle swarm algorithm initializes the process parameters as a population of random particles. Each particle represents a potential optimal solution to the extreme value optimization problem. In each iteration, the particle updates itself by tracking two “extremes”, and updates the individual extremum pbest and population extremum gbest by comparing the fitness value of the new particle with the individual extremum and the fitness value of the population extremum. During each iteration, the particle updates its velocity and position through individual and population extremes using Equations (10) and (11) [45].
(10)$$v_{i}^{d}=wv_{i}^{d-1}+c_{1}r_{1}\left(pbest_{i}^{d}-x_{i}^{d}\right)+c_{2}r_{2}\left(gbest^{d}-x_{i}^{d}\right)$$
(11)$$x_{i}^{d}=x_{i}^{d}+V_{i}^{d}$$
where $c_{1}$ is the individual acceleration factor; $c_{2}$ is the acceleration factor for society; $v_{i}^{d}$ is the d iteration velocity of the i rd particle; $pbest_{i}^{d}$ is until the d iteration the best position for the i st particle to pass through; $gbest^{d}$ is until the d iteration the best position for all particles to pass through.

Particle swarm optimization is very effective for many optimization problems. Much work has been carried out in this area [46]; therefore, it is possible to effectively control the amount of blade deformation. In this paper, the particle swarm algorithm is used to search for the optimal set of process parameters to obtain the minimum blade deformation. The particle swarm algorithm flow is shown in Figure 13.

### 4.3. Particle Swarm Algorithm Model Training

According to the flow chart of the particle swarm optimization algorithm, combined with the actual situation of blade milling processing, an optimization program was written in MATLAB to obtain the best set of solutions for Pbest and Gbest. The use of inertia weights w provides improved performance in many applications. Adaptive inertia weights $w_{min}$ = 0.4 and $w_{max}$ = 0.9 are chosen for this. Set the number of particles to 1000, and the four variables range from [1000, 3000], [0.05, 0.2], [0.5, 2] and [−196 °C, 20 °C], respectively; repeat the optimization five times for the particle swarm algorithm. The number of iterations is 100 so the fitness function is at a stable value, as shown in Figure 14.

In Figure 14, the corresponding fitness function value tends to 0 when the graph tends to stabilize after each number of iterations. There is good convergence of the optimized objective function values. However, due to the randomness in the search process of the particle swarm algorithm for finding the best value, each optimization result is different. Three sets of optimal fitness values were chosen as the corresponding variable parameters [1167, 0.2, 0.9, −164] [1291, 0.15, 1.6501, −181] [1110, 0.169, 0.8708, −156]. These results were used to simulate a milling deformation model using DEFORM and ABAQUS to verify the feasibility of the particle swarm algorithm optimization results.

### 4.4. Particle Swarm Algorithm Optimization and Validation

To verify the feasibility of the optimized data, the optimized three sets of process parameters are substituted back into the finite element simulation model, and the optimization data and simulation results are shown in Table 13.

As shown in the table, the milling parameters optimized by the particle swarm algorithm are the machining conditions. The maximum machining deformation of the blade is 22.90 µm and the maximum deformation of the blade under the simulated milling process parameters is 39.60 µm. The error value is 42.17%, and the average error is 43.89%. Therefore, the optimized process parameters have achieved the expected effect of improving the machining quality and contour accuracy and controlling the machining deformation of titanium alloy aero-engine blade milling. This meets the blade profile deformation tolerance of ±50 µm.

## 5. Conclusions

A finite element simulation model for blade milling under low-temperature conditions was established based on research into the fundamental theory of metal cutting and the key technology of finite element milling simulation. The numerical simulation of blade milling was completed, obtaining the milling force and temperature through the analysis of the blade structure characteristics and milling error. ABAQUS 2020 software was utilized to simulate thermal-force loading on the blade, completing the calculation of the blade surface.

An analysis by a single-factor control scheme revealed a significant reduction in blade deformation under low-temperature milling conditions. However, the blade profile accuracy exceeds ±50 µm. To effectively control blade machining deformation, a Box–Behnken test scheme was designed for the analysis of the relationship between the process parameters (spindle speed, feed per tooth, depth of cut, and jet temperature) and the amount of blade deformation according to Design-Expert. A quadratic multinomial regression model was obtained by fitting multiple regressions to the experimental data. The particle swarm algorithm was used to obtain the process parameters corresponding to the smallest deformation of the blade. Substitute the optimized process parameters into the finite element model, and the maximum blade deformation is 0.0396 mm, with an average error between the optimized and validated values of 43.89%, meeting the blade profile deformation amount error (±50 µm). This approach can achieve ideal machining accuracy, surface quality, and milling efficiency.

The jet temperature functions as the independent variable in the optimization algorithm, serving as a reference for selecting process parameters during actual processing and enhancing the stability of the processing system.

## Figures and Tables

**Figure 1 materials-16-04072-f001:**
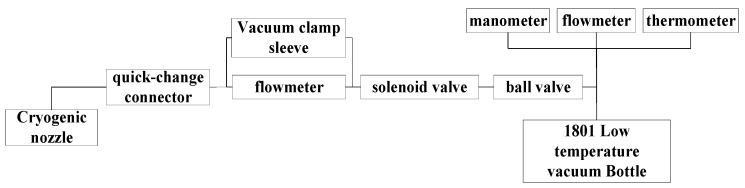
Layout diagram of the low-temperature jet system.

**Figure 2 materials-16-04072-f002:**
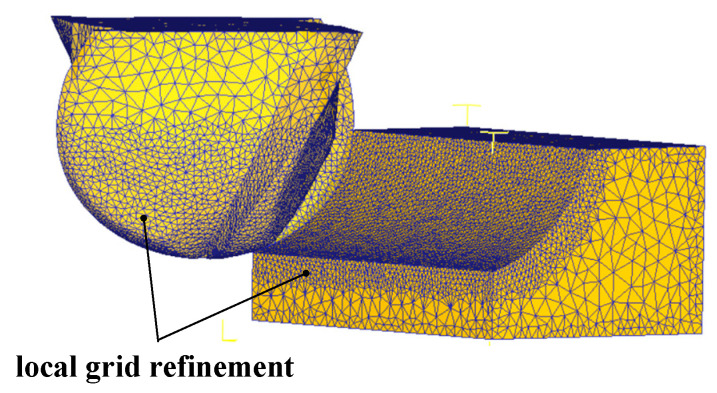
Cutter-workpiece model meshing diagram.

**Figure 3 materials-16-04072-f003:**
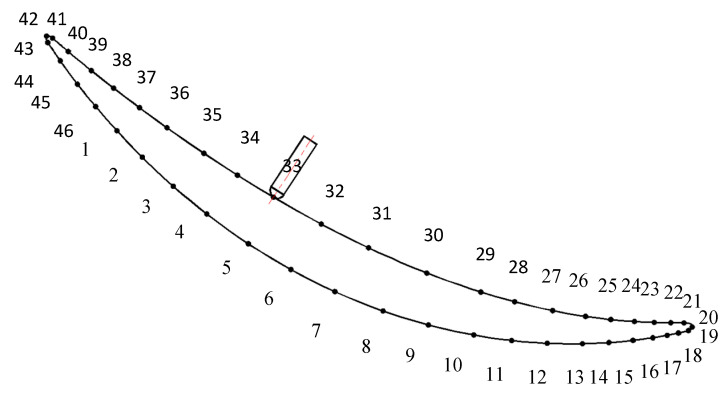
Blade milling path diagram.

**Figure 4 materials-16-04072-f004:**
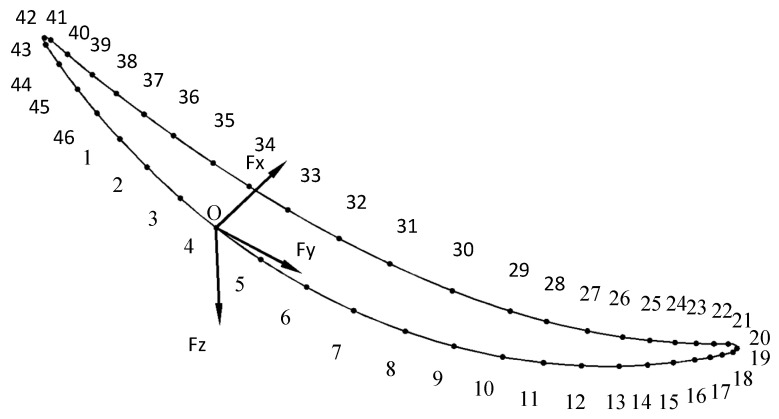
Blade section cutter contact numbering diagram. Figure Notes: O is the knife contact temperature boundary condition, F_x_, F_y_, and F_z_ are three-way milling forces.

**Figure 5 materials-16-04072-f005:**
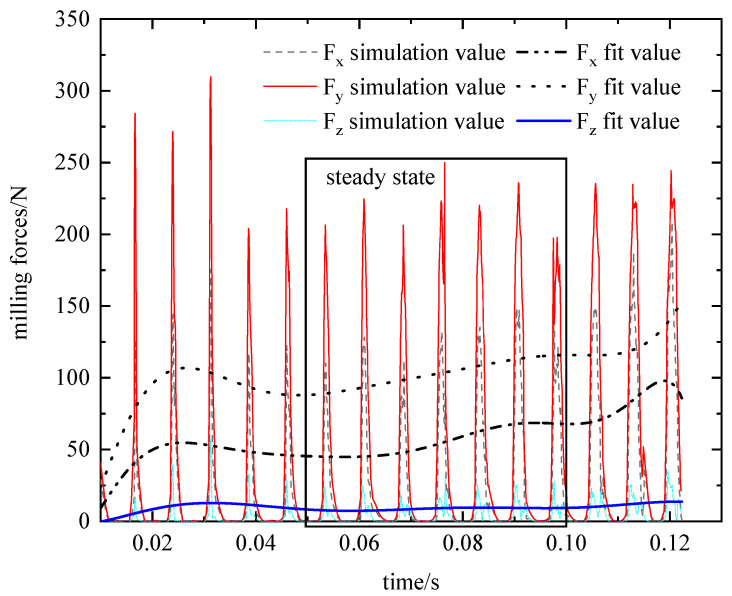
Milling force variation curve with machining time.

**Figure 6 materials-16-04072-f006:**
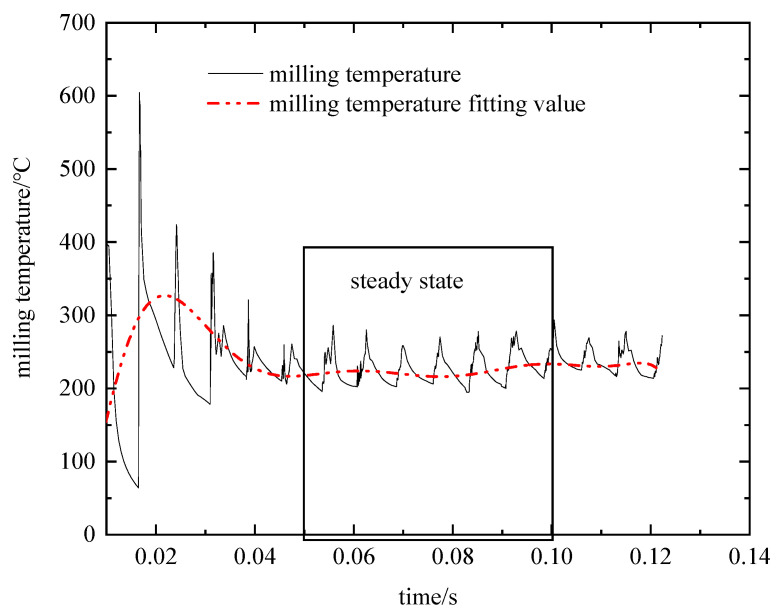
Milling temperature as a function of processing time.

**Figure 7 materials-16-04072-f007:**
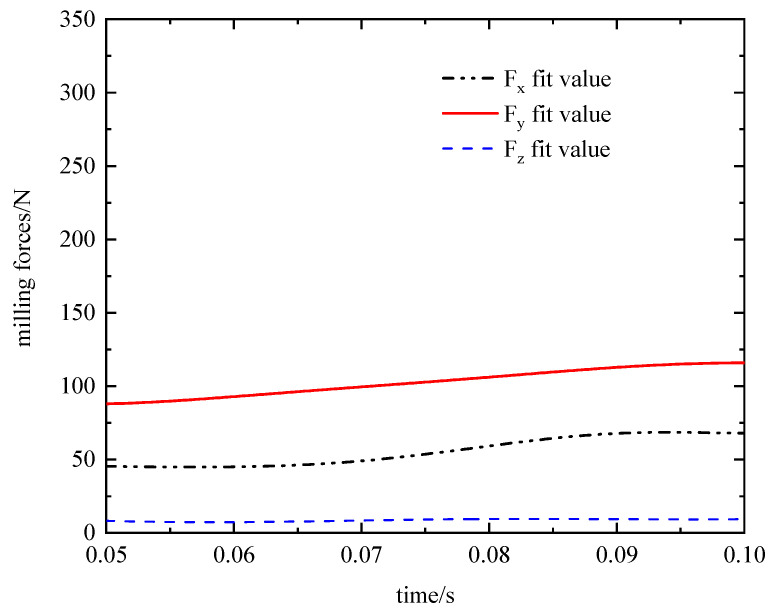
Milling force extraction diagram.

**Figure 8 materials-16-04072-f008:**
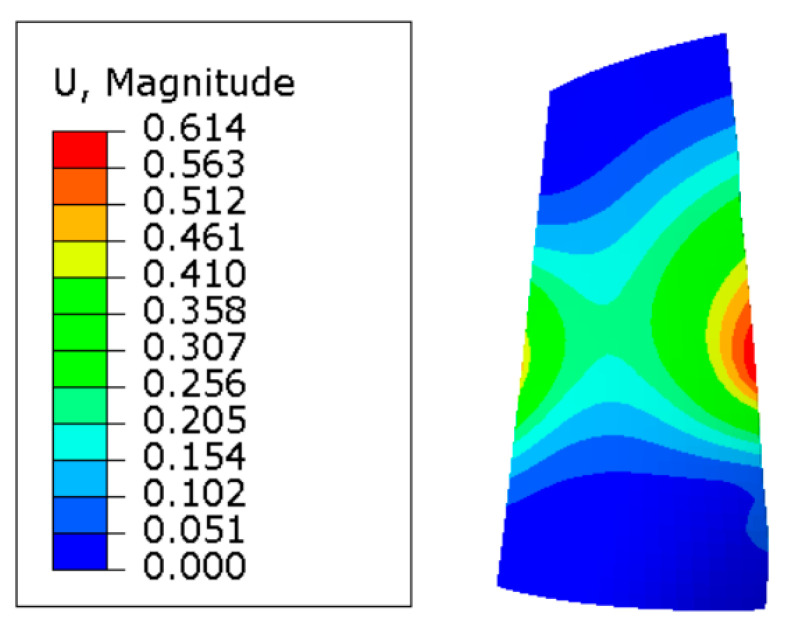
Cloud image of blade deformation.

**Figure 9 materials-16-04072-f009:**
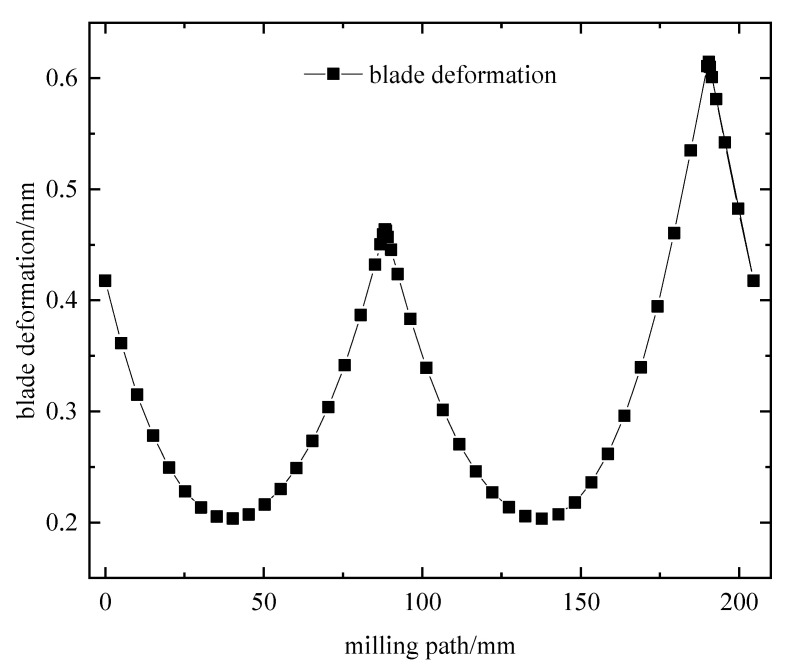
Graph of blade deformation and milling path.

**Figure 10 materials-16-04072-f010:**
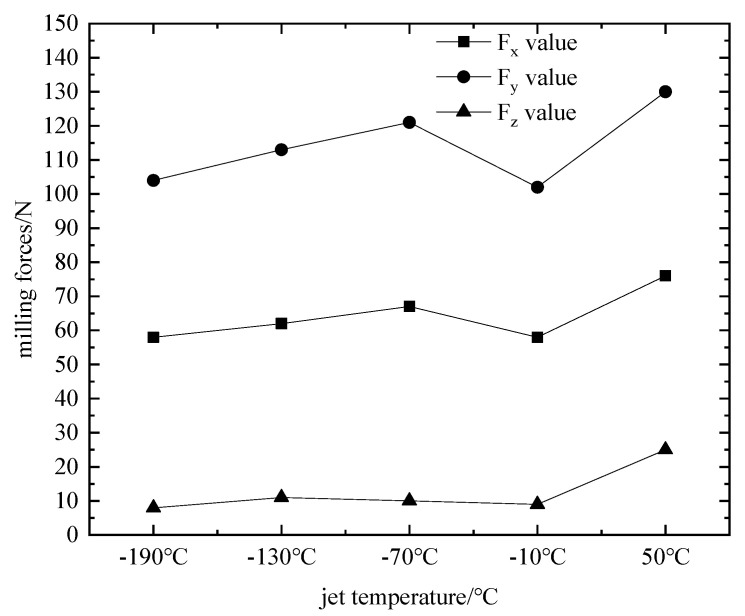
Effect of fluidic temperature on milling force.

**Figure 11 materials-16-04072-f011:**
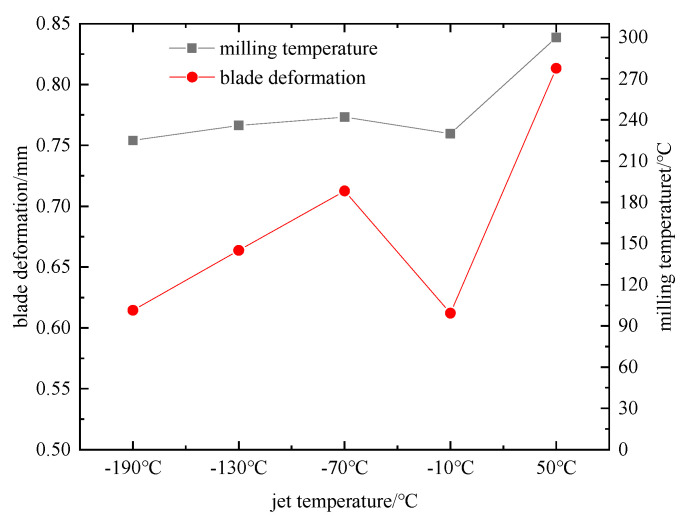
Effect of jet temperature on blade deformation and milling temperature.

**Figure 12 materials-16-04072-f012:**
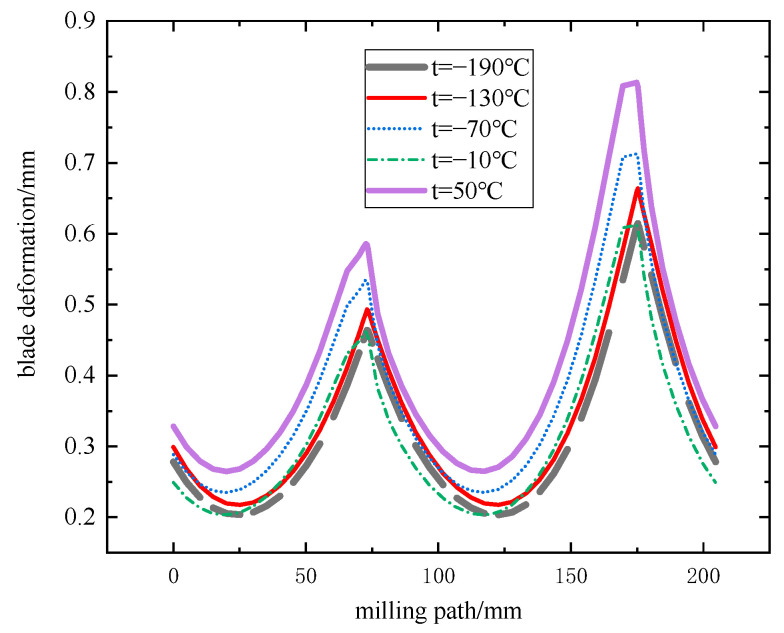
Graph of relationship between milling path and blade deformation.

**Figure 13 materials-16-04072-f013:**
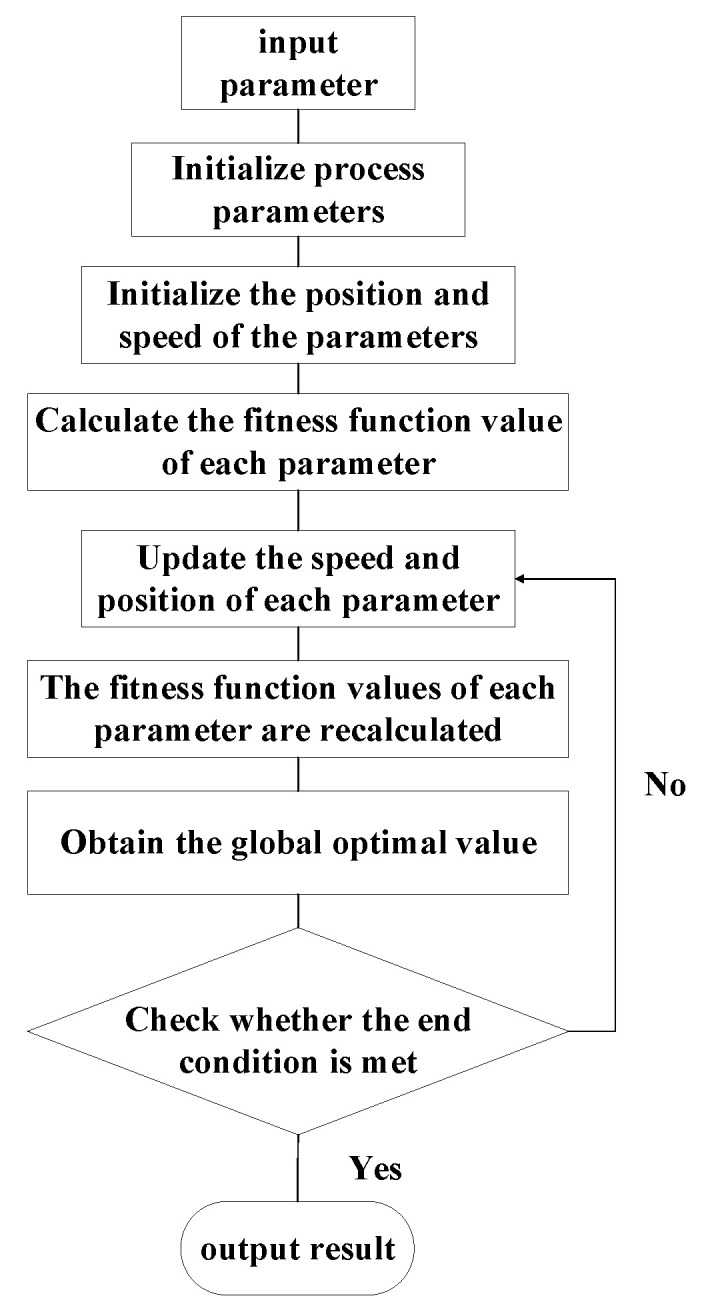
Flow chart of particle swarm optimization algorithm.

**Figure 14 materials-16-04072-f014:**
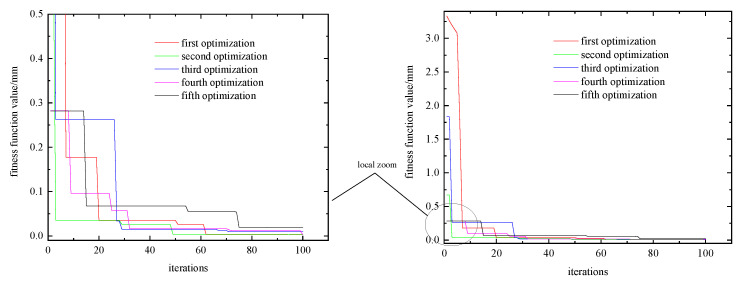
The number of iterations and the change in fitness function value.

**Table 1 materials-16-04072-t001:** Geometric parameters of ball end milling cutter.

Number of Edges	Anterior Angle $\gamma_{0}$ (°)	Relief Angle $a_{0}$ (°)	Helical Angle $\beta_{0}$ (°)	Diameter d (mm)	Blade Radius $r_{0}$ (mm)
4	15	15	30	10	0.02

**Table 2 materials-16-04072-t002:** Physical parameter.

Elasticity Modulus (GPa)	Poisson Ratio	Specific Heat (J/g °C)	Thermal Conductivity (W/m K)	Hardness (HRC)
650	0.25	0.96	59	70

**Table 3 materials-16-04072-t003:** Low-temperature TC4 physical parameters.

Elasticity Modulus (GPa)	Poisson Ratio	Heat Conductivity Coefficient (W/m K)	Specific Heat (J/kg K)	Melting Point K	Failure Displacement mm
10.79	0.33	2.22	2100	273	0.04

**Table 4 materials-16-04072-t004:** Johnson–Cook model parameters.

Materials	*A* (Mpa)	*B* (MPa)	*n*	*c*	*m*
TC4	861	331	0.34	0.03	0.8

**Table 5 materials-16-04072-t005:** Temperature dependence of low-temperature window HTC [41].

Temperature/°C	−195	−150	−100	−50	0	1600
HTC low-temperature jet window/($W/m^{2}K$)	0	280,000	100,000	30,000	20,000	20,000

**Table 6 materials-16-04072-t006:** Primary process parameter.

Speed of Main Shaft n/(r/min)	Feed Engagement $f_{z}$/(mm/z)	Knifing $a_{p}$/ (mm)	Jet Temperature t/ (°C)
2000	0.125	1.25	−190

**Table 7 materials-16-04072-t007:** Numerical simulation of the main performance parameters.

Simulated Total Number of Steps/Step	Incremental Step/(s/Step)	Friction Coefficient/(W/(m^2^ K))		Photoconductivity/(W/(m^2^ K))	
		Dry Milling	Low-Temperature Milling	Dry Milling	Low-Temperature Milling
800	0.0005	0.4	0.25	500	1500

**Table 8 materials-16-04072-t008:** Three-way milling force and milling temperature take point data table.

F_x_ Value/N	F_y_ Value/N	F_z_ Value/N	Milling Temperature *t*/°C
58	104	8	225

**Table 9 materials-16-04072-t009:** Single factor test of fluidic temperature variation.

Parameter	Group Indication				
	1	2	3	4	5
Temperature *t*/(°C)	−190°C	−130 °C	−70 °C	−10 °C	50 °C
Other parameters	n=2000 r/min		$f_{z}=0.125\;\mathrm{mm}/z$	$a_{p}=1.25\;\mathrm{mm}$	

**Table 10 materials-16-04072-t010:** Factor level table.

Test Level	Experimental Factor			
	A Spindle Speed/(r/min)	B Feed Per Tooth/(mm/z)	C Depth of Cut/(mm)	D Jet Temperature/(°C)
1.68	1000	0.050	0.50	−196
0	2000	0.125	1.25	−88
−1.68	3000	0.200	2.00	20

**Table 11 materials-16-04072-t011:** BBD test simulation results.

Group Number	Spindle Speed/(r/min)	Feed Per Tooth/(mm/z)	Depth of Cut/ mm	Jet Temperature/ °C	Blade Deformation /(µm)
1	2000	0.050	1.25	−196	46.60
2	2000	0.125	1.25	−88	254.00
3	2000	0.200	1.25	−196	622.90
4	2000	0.200	1.25	20	924.80
5	3000	0.125	2.00	−88	255.40
6	2000	0.125	1.25	−88	254.00
7	1000	0.050	1.25	−88	51.25
8	2000	0.050	0.50	−88	148.70
9	3000	0.050	1.25	−88	308.40
10	1000	0.125	0.50	−88	333.80
11	2000	0.125	1.25	−88	254.00
12	1000	0.125	1.25	20	890.00
13	2000	0.125	2.00	−196	546.30
14	2000	0.125	2.00	20	646.30
15	1000	0.125	2.00	−88	133.80
16	3000	0.200	1.25	−88	892.44
17	2000	0.050	2.00	−88	255.40
18	3000	0.125	0.50	−88	998.70
19	2000	0.200	0.50	−88	996.90
20	1000	0.125	1.25	−196	56.60
21	1000	0.200	1.25	−88	362.60
22	2000	0.125	0.50	20	1005.89
23	2000	0.125	1.25	−88	254.00
24	2000	0.125	0.50	−196	31.80
25	2000	0.050	1.25	20	563.80
26	2000	0.200	2.00	−88	892.29
27	3000	0.125	1.25	20	891.80
28	3000	0.125	1.25	−196	235.80
29	2000	0.125	1.25	−88	300.00

**Table 12 materials-16-04072-t012:** Quadratic model variance analysis.

Source	Quadratic Sum	Degree of Freedom	Mean Square	F Value	*p* Value	
Model	6.3500	14	0.4537	6.5400	0.0006	notable
A	0.2565	1	0.2565	3.7000	0.0750	
B	0.9173	1	0.9173	13.2300	0.0027	
C	0.4678	1	0.4678	6.7500	0.0211	
D	2.0500	1	2.0500	29.6300	<0.0001	
AB	0.0186	1	0.0186	0.2681	0.6127	
AC	0.0738	1	0.0738	1.0600	0.3198	
AD	0.0079	1	0.0079	0.1135	0.7412	
BC	0.0112	1	0.0112	0.1610	0.6943	
BD	0.0116	1	0.0116	0.1671	0.6889	
CD	1.5100	1	1.5100	21.7700	0.0004	
A^2^	0.0041	1	0.0041	0.0589	0.8118	
B^2^	0.0210	1	0.0210	0.3023	0.5911	
C^2^	0.4982	1	0.4982	7.1800	0.0179	
D^2^	0.6007	1	0.6007	8.6600	0.0107	
Residual	0.9708	14	0.0693			
Lack of Fit	0.9691	10	0.0969	229.00	<0.0001	notable
Pure Error	0.0017	4	0.0004			
Cor Total	7.3200	28				

**Table 13 materials-16-04072-t013:** Algorithm optimization and simulation verify blade deformation.

Group Number	Spindle Speed /mm	Feed Per Tooth /(mm/z)	Depth of Cut /mm	Jet Temperature/°C	Optimize the Amount of Deformation/µm	Simulation Deformation Amount/µm	Deviation /%	Average Deviation /%
1	1167	0.200	0.9000	−164	18.00	30.60	41.18%	43.89%
2	1291	0.150	1.6501	−181	22.90	39.60	42.17%	
3	1110	0.169	0.8708	−156	15.40	29.80	48.32%	

## Data Availability

The data presented in this study are available on request from the corresponding author.

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
