# Peer review of "Aero-Engine Blade Cryogenic Cooling Milling Deformation Simulation and Process Parameter Optimization"

_materials, 2023, doi:10.3390/ma16114072_

Round 1
Reviewer 1 Report
Introduction, Page 1, line 37: check missformatting;
Introduction, Pag 2, line 53: check misspelling;
Introduction: Most of the text refers to results attained by oyther authors. So, it should be improved in order to present the effects of cryogenic cooling on workpiece and tool material's properties;
Material and Methods, page 3, line 101: explain better "a" factor on equation (1);
Material and Methods, page 4: improve quality of figures 3 and 4;
Material and Methods, page 5: missformatting of Table 5;
Page 6, line 192: From this point on, is the data presented results?
Page 9: missformatting of Figure 5 label;
Page 11: improve Figure 13 in order to allow adequate comparison between deformations under different temperatures;
Page 11: Explain better Box-Benhnken test factors selection process;
Page 17: Conclusions must be clean and objective.
Experimental tests were conducted or just simulations on Abaqus?
Minor revisions should be taken.
Author Response
Dear Reviewers:
Thank you for your comments concerning our manuscript entitled “Aero-engine blade cryogenic cooling milling deformation simulation and process parameter optimization” (materials-2376087). Those comments are all valuable and very helpful for revising and improving our paper, as well as the important guiding significance to our researches. We have studied comments carefully and have made correction which we hope meet with approval. Revised portion are marked in red in the paper. The main corrections in the paper and the responds to the reviewer’s comments are as flowing:
1) Introduction, Page 1, line 37: check missformatting;
Response to comment: We sincerely thank the reviewer for careful reading. As suggested by the reviewer, we have corrected the “such as” into “For example”
2) Introduction, Pag 2, line 53: check misspelling;
Response to comment: We sincerely thank the reviewer for careful reading. As suggested by the reviewer, we have corrected the “PCB” into “PCBN”
3) Introduction: Most of the text refers to results attained by oyther authors. So, it should be improved in order to present the effects of cryogenic cooling on workpiece and tool material's properties;
Response to comment: As suggested by the reviewer, we have added more references to support this idea. “Shokrani A et al [14] After conducting drilling tests on carbon composites, it was discovered that liquid nitrogen ultra-low temperature machining can effectively reduce fiber delamination at the drill exit compared to dry cutting. Additionally, there was a 25% reduction in surface roughness after machining and an improvement in surface integrity. Fengbiao W [15] A series of ultra-low temperature internal cold milling tests were conducted on TC4 titanium alloy material. It was found that compared to conventional cutting fluid cooling, the ultra-low temperature cooling process resulted in regular sawtooth-shaped chips, suppressed adiabatic shear bands, and reduced microscopic defects in the material.”
4) Material and Methods, page 3, line 101: explain better "a" factor on equation (1);
Response to comment: In equation (1) the a is the expansion angle.
5) Material and Methods, page 4: improve quality of figures 3 and 4;
Response to comment: We have revised this part according to the Reviewer’s suggestion.
Figure 3. Blade milling path diagram.
Figure 4. Blade section cutter contact numbering diagram.
6) Material and Methods, page 5: missformatting of Table 5;
Response to comment: We have revised this part according to the Reviewer’s suggestion.
Table 5. Temperature dependence of low-temperature window HTC[26].
|
temperature/℃ |
-195 |
-150 |
-100 |
-50 |
0 |
1600 |
|
HTC Low-temperature Jet window /() |
0 |
280000 |
100000 |
30000 |
20000 |
20000 |
7) Page 6, line 192: From this point on, is the data presented results?
Response to comment: We have revised this part according to the Reviewer’s suggestion.
When contact is in the forced sticking region, where the tool is rigid, heat exchange occurs between the blade and tool, as shown in Eq 5. According to the heat distribution coefficient (HPC), , Heat flow sharing for tools, and heat flow shared for the workpiece (see Eq 6), the heat flow due to friction can be calculated from the sliding speed (see Eq 7). When calculating HPC, the accumulation of fluid can be based on every single object, where is the density, is the thermal conductivity, is the specific heat capacity. In this paper [27],
8) Page 9: missformatting of Figure 5 label;
Response to comment: We have revised this part according to the Reviewer’s suggestion.
9) Page 11: improve Figure 13 in order to allow adequate comparison between deformations under different temperatures;
Response to comment: We have revised this part according to the Reviewer’s suggestion.
10) Page 11: Explain better Box-Benhnken test factors selection process;
Response to comment: We sincerely thank the reviewer for careful reading. As suggested by the reviewer, we have corrected the “According to the actual working conditions, four factors were considered in this study, The spindle speed , feed per tooth , depth of cut , and jet temperature are used to construct the finite element model for blade milling deformation. To reduce the number of tests and improve the efficiency of test operations, the Box-Benhnken test scheme was used, the influence law of process parameters on blade deformation was studied, and the optimal combination of factor levels was obtained by ANOVA. The result is a workpiece with high machining accuracy, excellent surface quality, and high machining efficiency. The level values of each factor were obtained based on processing experience and experimental processing conditions, and are presented in Table 10.” into “There are many process parameters that affect blade machining deformation, including spindle speed, milling speed, milling depth, milling width, feed per tooth, tool rotation angle, and more. Each parameter has a wide range of values and the full-scale test method requires a large workload. The Box-Benhnken test scheme can achieve the same results as the full-scale test method with fewer tests which improves testing efficiency.
This study took four elements into account based on actual operating conditions: spindle speed , feed per tooth , depth of cut , and jet temperature . Each factor is assigned three levels. The influence law of process parameters on blade deformation was explored using the Box-Benhnken test, and the ideal combination of factor values was determined using ANOVA. As a result, the workpiece has outstanding surface quality, high machining accuracy, and high machining efficiency. Table 10 shows the level values for each element depending on processing experience and experimental processing settings.”
11) Page 17: Conclusions must be clean and objective.
Response to comment: We have revised this part according to the Reviewer’s suggestion.
A finite element simulation model for blade milling under low-temperature conditions was established based on research into the fundamental theory of metal cutting and the key technology of finite element milling simulation. The numerical simulation of blade milling was completed, obtaining the milling force and temperature through analysis of the blade structure characteristics and milling error. ABAQUS software was utilized to simulate thermal-force loading on the blade, completing the calculation of the blade surface.
Analysis by a single-factor control scheme, a significant reduction in blade deformation under low-temperature milling conditions. But the blade profile accuracy exceeds ±50µm.To effectively control blade machining deformation, a Box-Behnken test scheme was designed to the analysis of the relationship between the process parameters (spindle speed, feed per tooth, depth of cut, and jet temperature) and the amount of blade deformation according to Design-Expert. A quadratic multinomial regression model was obtained by fitting multiple regression to the experimental data. The particle swarm algorithm was used to obtain the process parameters corresponding to the smallest deformation of the blade. Substitute the optimized process parameters into the finite element model, and the Maximum blade deformation is 0.0396mm, with an average error between the optimized and validated values of 43.89%, meeting the blade profile deformation amount error (±50µm). This approach can achieve ideal machining accuracy, surface quality, and milling efficiency.
The jet temperature functions as the independent variable in the optimization algorithm, serving as a reference for selecting process parameters during actual processing and enhancing the stability of the processing system.
12) Experimental tests were conducted or just simulations on Abaqus?
Response to comment: experimental just simulations on abaqus and deform. if actual processing experiments are required to validate the accuracy of the simulation analysis, special liquid nitrogen ultra-low temperature CNC machine tools, ultra-low temperature liquid nitrogen refrigeration devices, special tools, and other supporting facilities are required. The financial investment required is significant. According to the information gathered, only Shanghai Aerospace Equipment Manufacturing General Factory and the Dalian University of Technology have conducted a certain range of experimental validation after receiving project funding support from the People's Republic of China's Ministry of Industry and Information Technology for major scientific and technological research projects in previous years. However, there will be no large-scale implementation. Only a few other domestic firms and research institutes are now primarily engaged in verification, there are fewer corresponding papers. As a result, in the current state and conditions, conduct simulation and analysis of various approaches. Theoretically, it is critical to investigate ultra-low temperature liquid nitrogen processing technology study.
We tried our best to improve the manuscript and made some changes marked in red in the revised paper which will not influence the content and framework of the paper. We appreciate for Reviewers’ warm work earnestly and hope the correction will meet with approval. Once again, thank you very much for your comments and suggestions.
All the best to your work
Have a nice day.
Best regards,
Ting Chen

Reviewer 2 Report
This article is about blade cryogenic cooling milling simulations which is an interesting theme. However, in my opinion, still needs a significant improvement before publishing.
1) There are some formatting issues throughout the document. There are missing spaces, uppercase words in the middle of sentences, and the legend at the beginning is in lowercase. Additionally, there are inconsistencies in spacing and font types throughout the manuscript. These issues are particularly noticeable in the equations, such as lines 183-192. I recommend addressing these formatting issues.
2)The sentence from 70-73 must be rephrased as is difficult to understand.
3) The introduction needs some improvement in connecting and relating the different mentioned works.
4) In equation (1) the a must be defined.
5) Figure 2 should be bigger to see the mesh.
6) All the parameters referred to in tables should be defined or explained in the text.
7) In equation (5), besides the format problems, if the e_t is defined, why it is not used in the equation to make it simpler?
8) I think Figure 8 is not necessary, it is enough to say that the milling temperature was kept stable over time.
9) Figure 10 should be bigger. How can you explain the oscillation in blade deformation?
10) Why is there a significant reduction in blade deformation at the point between -70oC and 50oC? Also, you could state the corresponding point temperatures more clearly.
11) Figure 13 could be maybe with the same magnitude to be easier to compare.
I think Tables 11 and 12 states the same, with the addition of blade deformation (r/min), so I suggest maintaining only Table 12 to avoid repetition.
12) In tables and calculations all the presented data should have the same number of significant figures (be consistent), and, if possible, present the associated deviation.
13) Equation 9 needs formatting improvements.
14) Regarding the flowchart in Figure 14, it may be helpful to change the wording of one of the options presented. Specifically, rather than using "If" as the alternate choice to "Yes" at the bifurcation point, it may be more appropriate to use "No".
15) I suggest adding a zoom of the section 0-20 iterations and 0-0.5 fitness function value of Figure 15 to clarify.
16) You could provide further explanation for the phrases "The algorithm optimizes the amount of deformation" and "The deformation is verified by simulation" in Table 14. These phrases should be replaced with more concise words in the table.
Author Response
Dear Reviewers:
Thank you for your comments concerning our manuscript entitled “Aero-engine blade cryogenic cooling milling deformation simulation and process parameter optimization” (materials-2376087). Those comments are all valuable and very helpful for revising and improving our paper, as well as the important guiding significance to our researches. We have studied comments carefully and have made correction which we hope meet with approval. Revised portion are marked in red in the paper. The main corrections in the paper and the responds to the reviewer’s comments are as flowing:
1) There are some formatting issues throughout the document. There are missing spaces, uppercase words in the middle of sentences, and the legend at the beginning is in lowercase. Additionally, there are inconsistencies in spacing and font types throughout the manuscript. These issues are particularly noticeable in the equations, such as lines 183-192. I recommend addressing these formatting issues.
Response to comment: We sincerely thank the reviewer for careful reading. We feel sorry for our carelessness. In our resubmitted manuscript, the Errors section is revised. Thanks for your correction.
2)The sentence from 70-73 must be rephrased as is difficult to understand.
Response to comment: We sincerely thank the reviewer for careful reading. As suggested by the reviewer, we have corrected the “Liu J Y et al [13]The cutting force, cutting temperature, and tool life of HSS in drilling nickel-based alloys was studied, as the Response surface method based on equivalent tool life, Optimization of cutting parameters for HSS drilling of nickel-based alloys, Then recommended the reasonable cutting parameters under the condition of using cutting fluid.” into “Liu J Y et al [13] The cutting force, cutting temperature, and tool life of high-speed steels in drilling Nickel-based superalloy Inconel 718 was investigated through orthogonal cutting experiments. Then the empirical method of multi-factor line regression was used to establish the empirical models of drilling torque, thrust force, cutting temperature, and tool life of the drills with emulsion as the cutting fluid, and to verify the validity of the established model, and the significance of the regression method is tested to verify the effectiveness of the model”
3)The introduction needs some improvement in connecting and relating the different mentioned works.
Response to comment: As suggested by the reviewer, we have added more references to support this idea. “Caudill J et al. [14] A three-dimensional finite element model for milling Ti6Al4V under low temperature and cutting fluid environment was developed based on Deform, Predicted cutting forces and temperature fields during machining, Results showing that milling at low temperatures can extend tool life and improve machining efficiency; Boulila A et al. [15] according to AISI 304 stainless steel blade milling process, study the effects of the cutting parameters on tool deflection, tracking errors, and surface roughness, results have shown how much the tool deflection and the tracking errors were mainly influenced by the axial depth and the cutting speed.”
4)In equation (1) the a must be defined.
Response to comment: In equation (1) the a is the expansion angle.
5)Figure 2 should be bigger to see the mesh.
Response to comment: We have revised this part according to the Reviewer’s suggestion.
6)All the parameters referred to in tables should be defined or explained in the text.
Response to comment: We have revised this part according to the Reviewer’s suggestion.
7)In equation (5), besides the format problems, if the e_t is defined, why it is not used in the
the equation to make it simpler?
Response to comment: We sincerely thank the reviewer for careful reading. As suggested by the reviewer, we have corrected the “” into “”
8)I think Figure 8 is not necessary, it is enough to say that the milling temperature was kept stable over time.
Response to comment: We have rewritten this part according to the Reviewer’s suggestion. Figure 8 was deleted.
9)Figure 10 should be bigger. How can you explain the oscillation in blade deformation?
Response to comment: We sincerely thank the reviewer for careful reading. As suggested by the reviewer, we have corrected the “From Fig. 9 and Fig. 10, it can be found that during the machining process with the feed of the tool the blade section is locally deformed, the Milling path of 190.4mm in the test corresponds to a maximum blade deformation of 0.6145mm, and the deformation position is near the intake and exhaust side on the blade. So, this paper selects the milling path of 190.4mm near the middle maximum for studying blade machining deformation control and process parameters optimization.” into “According to Fig. 8, the displacement of blade node coalescence is greatest in the middle of the leaf shape, gradually decreasing distortion from the center to both ends. This is due to the blade ends being fully constrained, the displacement is zero, the intake and exhaust sides being in the overhang position, and the thickness of the curved edge of the leaf edge being small in comparison to the thickness of the middle position, making it susceptible to deformation due to three-way milling force and milling heat. Fig 9 illustrates that during machining with the tool feed (load moving action), the amount of blade cross-section local tool contact deformation displays a "W" shape law. In the test, the milling path of 190.4mm equates to a maximum blade deformation of 0.6145mm. According to the research, the inlet and exhaust sides of the blades are more prone to deformation than the middle of the back and inner airway arcs. So, this paper selects the milling path of 190.4mm near the middle maximum for studying blade machining deformation control and process parameters optimization.
”
10)Why is there a significant reduction in blade deformation at the point between -70oC and 50oC? Also, you could state the corresponding point temperatures more clearly.
Response to comment: We sincerely thank the reviewer for careful reading. As suggested by the reviewer, we have corrected the “From the above trends in Fig 11, Fig 12, and Fig 13, the trend of increasing milling force and milling temperature in three directions at 50°C is obvious in the single-factor test. The slope of Fy with respect to Fx and Fz is maximum at this temperature. This can be attributed to heat transfer from the tool resulting in lower temperatures during the milling process and an increase in titanium alloy flow stress, which strengthens the material and reduces damage. Within the range of -190℃~-10℃, Fx, Fy, Fz, and milling temperature have a small range. However, there is a continuous trend of increasing with the jet temperature. Moreover, the blade deformation trend is always consistent with the trend of milling force and milling temperature in three directions. Blade deformation reaches 0.8134mm at 50℃, while the minimum blade deformation is 0.6121mm, and the Jet temperature of -10℃. The test proves: -190℃ is not the best temperature for controlling the deformation of the blade. Although the range of blade deformation variation is reduced relative to dry milling, the blade profile accuracy exceeds 50µm. Therefore, using low-temperature processing, the scrap rate is still high. In this paper, the particle swarm algorithm is used to optimize for this problem and improve blade machining accuracy.” into “From the above trends in Fig 10, Fig 11, and Fig 12, the trend of increasing milling force and milling temperature in three directions at 50°C is obvious in the single-factor test(Fx=76N、Fy=130N、Fz=25N、Milling temperature 300℃). The slope of Fy with respect to Fx and Fz is maximum at this temperature. This can be attributed to heat transfer from the tool resulting in lower temperatures during the milling process and an increase in titanium alloy flow stress, which strengthens the material and reduces damage. Within the range of -190℃~-10℃, Fx, Fy, Fz, and milling temperature have a small range. However, there is a continuous trend of increasing with the jet temperature. Moreover, the blade deformation trend is always consistent with the trend of milling force and milling temperature in three directions. Blade deformation reaches 0.8134mm at 50℃, while the minimum blade deformation is 0.6121mm, and the Jet temperature of -10℃. due to the small compressive strength at -70℃, the High frequency of fluctuation of milling force and milling temperature in the machining area, the maximum values are Fx=67N, Fy=121N, Fz=10N, and milling temperature 242℃ respectively, resulting in an abnormal increase in blade deformation. The test proves: -190℃ is not the best temperature for controlling the deformation of the blade. Although the range of blade deformation variation is reduced relative to dry milling, the blade profile accuracy exceeds 50µm. Therefore, using low-temperature processing, the scrap rate is still high. In this paper, the particle swarm algorithm is used to optimize for this problem and improve blade machining accuracy.”
11)Figure 13 could be maybe with the same magnitude to be easier to compare.I think Tables 11 and 12 states the same, with the addition of blade deformation (r/min), so I suggest maintaining only Table 12 to avoid repetition.
Response to comment: We have revised this part according to the Reviewer’s suggestion.
Figure 12. Cloud image of blade deformation.
12)In tables and calculations all the presented data should have the same number of significant figures (be consistent), and, if possible, present the associated deviation.
Response to comment: We have revised this part according to the Reviewer’s suggestion.
Table 11. BBD test simulation results.
|
Group number |
spindle speed /(r/min) |
feed per tooth /(mm/z) |
depth of cut / mm |
jet temperature/ ℃ |
blade deformation /(µm) |
|
1 |
2000 |
0.05 |
1.25 |
-196 |
46.6 |
|
2 |
2000 |
0.125 |
1.25 |
-88 |
254 |
|
3 |
2000 |
0.2 |
1.25 |
-196 |
622.9 |
|
4 |
2000 |
0.2 |
1.25 |
20 |
924.8 |
|
5 |
3000 |
0.125 |
2 |
-88 |
255.4 |
|
6 |
2000 |
0.125 |
1.25 |
-88 |
254 |
|
7 |
1000 |
0.05 |
1.25 |
-88 |
51.25 |
|
8 |
2000 |
0.05 |
0.5 |
-88 |
148.7 |
|
9 |
3000 |
0.05 |
1.25 |
-88 |
308.4 |
|
10 |
1000 |
0.125 |
0.5 |
-88 |
333.8 |
|
11 |
2000 |
0.125 |
1.25 |
-88 |
254 |
|
12 |
1000 |
0.125 |
1.25 |
20 |
890 |
|
13 |
2000 |
0.125 |
2 |
-196 |
546.3 |
|
14 |
2000 |
0.125 |
2 |
20 |
646.3 |
|
15 |
1000 |
0.125 |
2 |
-88 |
133.8 |
|
16 |
3000 |
0.2 |
1.25 |
-88 |
892.44 |
|
17 |
2000 |
0.05 |
2 |
-88 |
255.4 |
|
18 |
3000 |
0.125 |
0.5 |
-88 |
998.7 |
|
19 |
2000 |
0.2 |
0.5 |
-88 |
996.9 |
|
20 |
1000 |
0.125 |
1.25 |
-196 |
56.6 |
|
21 |
1000 |
0.2 |
1.25 |
-88 |
362.6 |
|
22 |
2000 |
0.125 |
0.5 |
20 |
1005.89 |
|
23 |
2000 |
0.125 |
1.25 |
-88 |
254 |
|
24 |
2000 |
0.125 |
0.5 |
-196 |
31.80 |
|
25 |
2000 |
0.05 |
1.25 |
20 |
563.8 |
|
26 |
2000 |
0.2 |
2 |
-88 |
892.29 |
|
27 |
3000 |
0.125 |
1.25 |
20 |
891.8 |
|
28 |
3000 |
0.125 |
1.25 |
-196 |
235.8 |
|
29 |
2000 |
0.125 |
1.25 |
-88 |
300 |
13)Equation 9 needs formatting improvements.
Response to comment: We sincerely thank the reviewer for careful reading. As suggested by the reviewer, we have corrected the “” into “”
14) Regarding the flowchart in Figure 14, it may be helpful to change the wording of one of the options presented. Specifically, rather than using "If" as the alternate choice to "Yes" at the bifurcation point, it may be more appropriate to use "No".
Response to comment: We have revised this part according to the Reviewer’s suggestion.
15)I suggest adding a zoom of the section 0-20 iterations and 0-0.5 fitness function value of Figure 15 to clarify.
Response to comment: We have revised this part according to the Reviewer’s suggestion.
16)You could provide further explanation for the phrases "The algorithm optimizes the amount of deformation" and "The deformation is verified by simulation" in Table 14. These phrases should be replaced with more concise words in the table.
Response to comment: We have revised this part according to the Reviewer’s suggestion.
Table 13. Algorithm optimization and simulation verify blade deformation.
|
Group number |
spindle speed /mm |
feed per tooth /(mm/z) |
depth of cut /mm |
jet temperature /℃ |
Optimize the amount of deformation/µm |
Simulation deformation amount/µm |
deviation /% |
average deviation /% |
|
1 |
1167 |
0.2 |
0.9 |
-164 |
18 |
30.6 |
41.18% |
43.89% |
|
2 |
1291 |
0.15 |
1.6501 |
-181 |
22.9 |
39.6 |
42.17% |
|
|
3 |
1110 |
0.169 |
0.8708 |
-156 |
15.4 |
29.8 |
48.32% |
We tried our best to improve the manuscript and made some changes marked in red in the revised paper which will not influence the content and framework of the paper. We appreciate for Reviewers’ warm work earnestly and hope the correction will meet with approval. Once again, thank you very much for your comments and suggestions.

Reviewer 3 Report
This work is purely simulation, which is not suitable for acceptance without actual cutting experimental validation. Hence, the authors are highly recommended to validate the authenticity of their simulation with actual cutting results.
English is fine.
Author Response
Dear Reviewers:
Thank you for your comments concerning our manuscript entitled “Aero-engine blade cryogenic cooling milling deformation simulation and process parameter optimization” (materials-2376087).
if actual processing experiments are required to validate the accuracy of the simulation analysis, special liquid nitrogen ultra-low temperature CNC machine tools, ultra-low temperature liquid nitrogen refrigeration devices, special tools, and other supporting facilities are required. The financial investment required is significant. According to the information gathered, only Shanghai Aerospace Equipment Manufacturing General Factory and the Dalian University of Technology have conducted a certain range of experimental validation after receiving project funding support from the People's Republic of China's Ministry of Industry and Information Technology for major scientific and technological research projects in previous years. However, there will be no large-scale implementation. Only a few other domestic firms and research institutes are now primarily engaged in verification, there are fewer corresponding papers. As a result, in the current state and conditions, conduct simulation and analysis of various approaches. Theoretically, it is critical to investigate ultra-low temperature liquid nitrogen processing technology study.
We appreciate for Reviewers’ warm work earnestly and hope the article will meet with approval. Once again, thank you very much for your comments and suggestions.
All the best to your work
Have a nice day.
Best regards,
Ting Chen
Round 2
Reviewer 2 Report
I appreciate the reviews made on the paper. I am glad to see that the modifications have improved the work. However, there are still some issues with the formatting. For instance, some words have capital letters in the middle of sentences. It is also important that all numbers have the same number of decimal places. For example, in Table 11, some results have two decimal places, so all numbers referring to the same results should have two decimal places as well. This issue appears frequently and needs to be reviewed throughout the entire work.
Author Response
Dear Reviewers:
We would like to thank you for the opportunity to revise and resubmit our manuscript (materials-2376087) , entitled " Aero-engine blade cryogenic cooling milling deformation simulation and process parameter optimization ". We have addressed the comments raised by the reviewers, and the amendments are highlighted in red in the revised manuscript.The main corrections in the paper and the responds to the reviewer’s comments are as flowing:
1)some words have capital letters in the middle of sentences
Response to comment: We sincerely thank the reviewer for careful reading. We feel sorry for our carelessness. In our resubmitted manuscript, the Errors section is revised. Thanks for your correction.
2)all numbers have the same number of decimal places. For example, in Table 11, some results have two decimal places, so all numbers referring to the same results should have two decimal places as well
Response to comment: We have revised this part according to the Reviewer’s suggestion.
we hope that these modifications can fulfill the requirements to make the manuscript acceptable for publication. Please let us know if you have any concerns about the manuscript and we would like to address them as soon as possible. Thank you again for your consideration of the revised manuscript.
All the best to your work
Have a nice day.
Best regards,
Ting Chen

Reviewer 3 Report
The authors have applied both numerical & particle swarm optimization to identify the key processing parameters to achieve a minimum blade deformation in the simulation for cryogenic-assisted milling.
The paper has fully addressed the recommendations for improvement and is ready for publication, even though I personally suggested for actual experimental validations to convince their proof-of-concept.
Author Response
Dear Reviewers:
Thanks very much for your kind work and consideration of the publication of our paper.
All the best to your work
Have a nice day.
Best regards,
Ting Chen